# Integrating remote monitoring into heart failure patients' care regimen: A pilot study

Albert Sohn[1], William Speier[1], Esther Lan[2], Kymberly Aoki[2], Gregg C. Fonarow[3], Michael K. Ong[2], Corey W. Arnold[1,4]*

1 Department of Radiological Sciences, University of California, Los Angeles, Los Angeles, California, United States of America, 2 Division of General Internal Medicine & Health Services Research, Department of Medicine, University of California, Los Angeles, Los Angeles, California, United States of America, 3 Division of Cardiology, Department of Medicine, University of California, Los Angeles, Los Angeles, California, United States of America, 4 Department of Pathology & Laboratory Medicine, University of California, Los Angeles, Los Angeles, California, United States of America

* cwarnold@ucla.edu

**Data Availability Statement:** All relevant data are within the manuscript and its Supporting Information files.

**Funding:** This work was supported by the National Institutes of Health (https://www.nih.gov) and the

## Abstract

### Background

Around 50% of hospital readmissions due to heart failure are preventable, with lack of adherence to prescribed self-care as a driving factor. Remote tracking and reminders issued by mobile health devices could help to promote self-care, which could potentially reduce these readmissions.

### Objective

We sought to investigate two factors: (1) feasibility of enrolling heart failure patients in a remote monitoring regimen that uses wireless sensors and patient-reported outcome measures; and (2) their adherence to using the study devices and completing patient-reported outcome measures.

### Methods

Twenty heart failure patients participated in piloting a remote monitoring regimen. Data collection included: (1) physical activity using wrist-worn activity trackers; (2) body weight using bathroom scales; (3) medication adherence using smart pill bottles; and (4) patient -reported outcomes using patient-reported outcome measures.

### Results

We evaluated 150 hospitalized heart failure patients and enrolled 20 individuals. Two factors contributed to 50% (65/130) being excluded from the study: smartphone ownership and patient discharge. Over the course of the study, 60.0% of the subjects wore the activity tracker for at least 70% of the hours, and 45.0% used the scale for more than 70% of the days. The pill bottle was used less than 10% of the days by 55.0% of the subjects.

National Heart, Lung, and Blood Institute (https://www.nhlbi.nih.gov) under grant R56HL135425 awarded to C.W.A., PhD. The funders had no role in study design, data collection and analysis, decision to publish, or preparation of the manuscript.

**Competing interests:** GCF reports consulting for Abbott, Amgen, Bayer, Janssen, Medtronic, and Novartis.

## Conclusions

Our method of recruiting heart failure patients prior to hospital discharge may not be feasible as the enrollment rate was low. Once enrolled, the majority of subjects maintained a high adherence to wearing the activity tracker but low adherence to using the pill bottle and completing the follow-up surveys. Scale usage was fair, but it received positive reviews from most subjects. Given the observed usage and feedback, we suggest mobile health-driven interventions consider including an activity tracker and bathroom scale. We also recommend administering a shorter survey more regularly and through an easier interface.

## Introduction

### Background

In the United States, at least 6.2 million adults currently live with heart failure [1]. Heart failure prevalence is projected to increase by 46% between 2012 and 2030, resulting in more than 8 million adults with heart failure [1]. Physical inactivity, obesity, and smoking are well-known lifestyle risk factors for heart failure [2]. Guidelines for secondary prevention after diagnosis emphasize physical activity, weight management, smoking cessation, and medication adherence [3]. Despite advances in medications and guidelines for heart failure management, heart failure considerably increases the risk of morbidity and mortality. Incidence of, morbidity resulting from, and hospitalization due to heart failure have substantial financial implications. Total direct medical costs of heart failure are expected to rise from $21 billion to $53 billion between 2012 and 2030 [1]. Total direct and indirect costs are estimated to increase by 127% from $30.7 billion in 2012 to $69.7 billion in 2030 [1].

Hospitalization is common among heart failure patients and is a significant driver of heart failure-related costs. Annually, there are more than 4 million hospitalizations with a primary or secondary diagnosis of heart failure [4], and they constitute up to 79% of the costs for heart failure treatment [5]. In 2013, heart failure was the sixth most expensive condition treated in US hospitals as costs reached $10.2 billion [6], with readmissions accounting for $2.7 billion [7]. Among heart failure patients, 83% are hospitalized at least once, and 43% are hospitalized at least four times [8]. Within the first 30 days of discharge, hospital readmission rates for heart failure patients exceed 20% [9, 10], while they near 50% within 6 months of discharge [11]. However, as high as 75% of the 30-day readmissions may be preventable [12] by addressing the patients' lack of information, comprehension, or adherence to prescribed self-care [13, 14].

Health care expenditures for heart failure increases with an aging population, and thus preventing heart failure and improving care efficiency are imperative. Since weight change or medication non-adherence alone, for example, may not correlate with hospitalizations [15–17], past home monitoring interventions have utilized a variety of methods, including wireless sensors, telephone services, websites, and home visits from nurses [18–21]. Results of these interventions designed to improve health outcomes and reduce readmissions are inconclusive. Additionally, patient adherence to telemedicine interventions is often low [22]. The poor adherence and negative results are in part due to the high treatment burden home monitoring interventions place upon patients. In most cases, they require patients to engage in novel behaviors such as using unfamiliar hardware and following high-frequency manual measurement regimen (e.g., taking one's blood pressure or heart rate multiple times a day).

## Objective

Mobile health (mHealth), defined as the application of mobile technology (e.g., software apps on mobile devices, wireless sensors, etc.) in health care, may be a preferred minimally invasive alternative to telemedicine interventions [23, 24]. Activity trackers are examples of mHealth devices that have been studied due individuals' relatively high adherence to wearing them upon recommendation. A previous study using commercial activity trackers produced adherence rates that were as high as 90% [25]. There were two main factors we sought to learn more about in this pilot study: (1) is it feasible to enroll heart failure patients in a remote monitoring regimen that uses wireless sensors and patient-reported outcome (PRO) measures; and (2) once enrolled, how adherent would patients be in using the study devices and completing PRO measures. In this work, we investigate the feasibility of recruitment and potential of the remote monitoring regimen by detailing preliminary results while identifying real-world problems and solutions to enable larger future studies.

## Methods

### Ethics

Data collection and analysis presented in this work were carried out under research protocol #17–001312 approved by University of California, Los Angeles, IRB. We obtained signed informed consent from all participants in the study.

### Recruitment

All patients who were admitted as an inpatient or for observation from May 2018 to June 2019 were prescreened for inclusion in the study. Those who were 50 years of age or older and were being actively treated for heart failure were considered eligible for the study. Additional criteria included ownership of a compatible smartphone device (iOS or Android) with cellular voice and data, in addition to access to a Wi-Fi connection in their home. Eligible patients who were interested in the study had to score 3.5 or higher on a shortened version of the Mobile Device Proficiency Questionnaire (MDPQ) to enroll (S1 Table in S1 Appendix) [26]. Exclusion criteria included having a cognitive disability (e.g., dementia), being unable to communicate in English, having visual or auditory impairments to the extent that a smartphone could not be used, having a full-time caregiver, and enrollment or being in the process of enrolling in hospice. Prior to discharge and enrollment in the study, eligible patients signed an informed consent form, which described the baseline survey, follow-up surveys, and institutional review board-approved procedures. Subjects were not paid but were allowed to keep the study devices, which they were given at the beginning of the study after completing the baseline survey in person with a study coordinator.

Each subject's New York Heart Failure Association (NYHA) classification and ejection fraction (EF) were recorded to describe the patients' heart failure according to the severity of their symptoms and limitations. The NYHA classification categorizes the severity of heart failure by considering heart failure patients' symptoms at rest and during physical activity [27]. EF indicates the percentage of blood leaving the left ventricle when it contracts and is a measurement of the heart's degree of function.

### Remote monitoring regimen

Upon providing signed informed consent, subjects received a Fitbit Charge 2 (Fitbit, Inc., San Francisco, CA, USA), a bathroom scale (BodyTrace, Inc., Palo Alto, CA, USA), and a smart pill bottle. The Fitbit Charge 2 is a wrist-worn consumer product that uses a combination of

accelerometers and optical sensors to track activity, heart rate (HR), and sleep based on arm movement and wrist capillary oxygenation. Subjects were asked to wear the Fitbit activity tracker continuously (Table 1), with interruptions for only activities involving water and charging the device. Data synced to users' smartphones via the Fitbit app, where it was then uploaded to the Fitbit cloud database and subsequently pulled to the research server via Fitbit's application programming interfaces (APIs). The BodyTrace scale is a wireless bathroom scale that digitally captures weight. Weights were automatically uploaded to the BodyTrace cloud database via cellular modem data connection after every use (Table 1) and were available via an API. The smart pill bottle has a smart cap that automatically tracks its removal from the bottle. This signal conveys information on medication consumption and was sent to its companion smartphone app via Bluetooth (Table 1). Cap removal events were available via an API.

A web-based data integration platform was utilized to collect all data streams for remote monitoring. This platform used vendor APIs to retrieve data every hour from Fitbit and once per day from BodyTrace's and the smart pill bottle's APIs. If the Fitbit activity tracker did not sync for 48 hours, study personnel issued a text message to the subject followed by a phone call if it did not sync for 72 hours. If the bathroom scale or the smart pill bottle did not sync for 72 hours, the subject received a text message as well as a phone call after 96 hours. Subjects were allowed to opt out of these reminders.

Study IDs were used to identify subjects, and collected data were stored in a HIPAA-compliant encrypted database. Data recorded for each subject included the subject's contact information and study-specific information including discharge date and study completion date, as well as withdrawal date and expiration date, if applicable. Because some of the daily readings

**Table 1. mHealth for heart failure protocol data collection.**

| Data Collection | Baseline | Day 30 | Day 90 | Day 180 |
|---|---|---|---|---|
| Remote Monitoring | | | | |
| Fitbit Charge 2 | Continuous | | | |
| BodyTrace Bathroom Scale | Upon Usage | | | |
| Smart Pill Bottle | Upon Usage | | | |
| Post-Discharge Questionnaires | | | | |
| Rapid Estimate of Adult Literacy in Medicine (REALM-7) | ✓ | ✗ | ✗ | ✗ |
| Demographics | ✓ | ✗ | ✗ | ✗ |
| Health Information National Trends Survey (HINTS) | ✓ | ✗ | ✗ | ✗ |
| Minnesota Living with Heart Failure Questionnaire (MLHFQ) | ✓ | ✗ | ✗ | ✗ |
| Self-Care of Heart Failure Index (SCHFI) | ✓ | ✓ | ✓ | ✓ |
| Geriatric Depression Scale (GDS-15) | ✓ | ✗ | ✗ | ✗ |
| Lubben Social Network Scale (LSNS-6) | ✓ | ✗ | ✗ | ✗ |
| Seattle Angina Questionnaire (SAQ-7) | ✓ | ✓ | ✓ | ✓ |
| Kansas City Cardiomyopathy Questionnaire (KCCQ-12) | ✓ | ✓ | ✓ | ✓ |
| Patient-Reported Outcomes Measurement Information System (PROMIS) Global Health | ✓ | ✓ | ✓ | ✓ |
| PROMIS Physical Function Short Form (SF) | ✓ | ✓ | ✓ | ✓ |
| PROMIS Fatigue SF | ✓ | ✓ | ✓ | ✓ |
| PROMIS Anxiety SF | ✓ | ✓ | ✓ | ✓ |
| PROMIS Depression SF | ✓ | ✓ | ✓ | ✓ |
| PROMIS Sleep Disturbance SF | ✓ | ✓ | ✓ | ✓ |
| PROMIS Social Isolation SF | ✓ | ✓ | ✓ | ✓ |
| Hospital Readmission Questions | ✗ | ✓ | ✓ | ✓ |
| Emergency Room (ER) Questions | ✗ | ✓ | ✓ | ✓ |
| Device Questions | ✗ | ✓ | ✓ | ✓ |

may have ceased if a subject were hospitalized, study personnel monitored the university-based hospital system for hospital readmissions. When appropriate, readmission date(s) and readmission discharge date(s) were documented. Withdrawal of participation occurred if a subject requested withdrawal. In the event that a subject withdrew from the study or expired, study personnel stopped contacting the subject for follow-up surveys.

## Post-discharge surveys

Study personnel contacted each subject on four separate occasions to complete a total of four surveys. The first was administered during the enrollment process and served as the baseline. It consisted of 16 sections (Table 1), including one that encompassed questions about sociode-mographic characteristics (S2 Table in S1 Appendix). After 30 days, 90 days, and 180 days, a follow-up survey was administered via phone by a member of the study team. Study team members called subjects during a window that started three days before and ended three days after the aforementioned follow-up periods. If a subject could not be contacted during that time frame, a paper copy of the follow-up survey was mailed to the subject's home address along with a pre-addressed, postage-paid return envelope. The 30-day, 90-day, and 180-day follow-up surveys each consisted of 13 sections (Table 1), which included questions about hospital readmission(s), emergency room (ER) visit(s), and study devices (S3 Table in S1 Appendix).

The baseline survey included the Rapid Estimate of Adult Literacy in Medicine (REALM-7) to determine the subjects' health literacy [28] and the Health Information National Trends Survey (HINTS) to assess the subjects' experience with technology (Table 1) [29]. Subjects also completed the Minnesota Living with Heart Failure Questionnaire (MLHFQ), Self-Care of Heart Failure Index (SCHFI), Geriatric Depression Scale (GDS-15), Lubben Social Network Scale (LSNS-6), Seattle Angina Questionnaire (SAQ-7), Kansas City Cardiomyopathy Questionnaire (KCCQ-12), and seven different Patient-Reported Outcomes Measurement Information System (PROMIS) questionnaires (Table 1). The MLHFQ is a 21-item patient-oriented measurement of health-related quality of life (HRQOL) [30]. It measures how heart failure affects a patient in three specific areas: physical, emotional, and socioeconomic. The GDS-15 is a screening test for depression among elderly populations [31], while the LSNS-6 measures the strength of social support networks among elderly populations [32]. The SCHFI assesses a patient's ability to manage their heart failure via 22 total items in three subscales: maintenance, management, and confidence [33]. The SAQ-7 and KCCQ-12 questionnaires evaluate patients' HRQOL in regard to angina and heart failure, respectively [34, 35]. Lastly, the PROMIS questionnaires are publicly available individual-centered PRO measures [36, 37].

## Scoring

Physical (0–40) and emotional (0–25) scores for the MLHFQ were calculated by summation of corresponding responses. Lower scores signified better HRQOL with respect to physical and emotional well-being, while higher scores signified worse HRQOL [30]. Addition of all 21 responses generated a total score, creating a possible range of 0 to 105. The following represents the classification of scores: good (<24), moderate (24–45), and poor (>45) HRQOL [30]. For the GDS-15, each *yes or no* question had a designated answer that was indicative of depression [31]. The number of answers matching those indicative of depression was the total score. Scores ≤5 are not suggestive depression, whereas >5 suggests depression and ≥10 is almost always indicative of depression [31]. LSNS-6 total scores were derived by addition of corresponding responses. Each question was scored from 0 to 5, with less than monthly, none, and

always representing 0 and daily, nine or more, and always denoting 5 [32]. The total score has a range of 0 to 30, and scores ≤12 suggest at-risk for social isolation [32].

Raw scores for the three SCHFI subscales were determined by summation of responses in each section. If the subject acknowledged having trouble breathing or ankle swelling within the past month of taking this survey, only then was the management raw score calculated [33]. Standardization of the raw scores were performed to a 0 to 100 scale, with higher scores indicating better self-care. Adequate self-care is defined as scores ≥70 for all sections of the SCHFI [33]. Addition of all corresponding responses for both the SAQ-7 and KCCQ-12 questionnaires produced raw scores. They were then standardized to a 0 to 100 range. For scores of both questionnaires, they are classified as poor (0–24), fair (25–49), good (50–74), and excellent (75–100) HRQOL in regard to angina and heart failure, respectively [34, 35]. Raw scores for the PROMIS questionnaires were computed by summation of responses to each questionnaire. Raw scores were then converted to *t* scores through a process of standardizing the scores to a mean of 50 and a standard deviation (SD) of 10 [36, 37]. Function scores greater than or equal to 40 are considered normal, and scores less than 40 represent moderate to severe adverse health effects. Whereas for symptoms, scores less than or equal to 60 are considered normal, and scores greater than 60 denote moderate to severe adverse health effects [36, 37].

## Statistical analysis

Each questionnaire in the baseline and follow-up surveys was scored prior to statistical analyses. Missing items in questionnaires using Likert scales were substituted by the mean of the subject's responses from the same questionnaire. However, those missing more than 20% of the items were deemed incomplete and not considered, since reliability declines when missingness is >20% [38]. The cohort was characterized using proportions, means, SDs, medians, and interquartile ranges (IQRs). For each questionnaire, summaries of responses and scores were reported, if applicable.

Adherence to wearing the Fitbit Charge 2 was calculated using two methods: (1) HR by hour (HR-hour); and (2) HR by minute (HR-minute). We examined the data by the hour for HR-hour and by the minute for HR-minute. HR-hour is the number of hours in the study that each subject had a heartrate recording divided by the total number of hours in the study. Similarly, HR-minute is the number of minutes in the study that each subject had a heartrate recording divided by the total number of minutes in the study. In an attempt to distinguish the subject from others in the household for adherence to weight measurement, weight data for each subject were averaged, and their SDs were calculated. Only the weights within 3.5 SDs from each subject's mean weight were defined to represent subject usage. For both the bathroom scale and smart pill bottle, adherence was calculated by taking the number of days in the study that each subject had data recorded on the respective device and dividing it by the total number of days in the study.

Correlation analyses were conducted with subject characteristics (i.e., NYHA classification, EF, age, education, and annual income) to quantify their relationships with the subjects' adherence to using the study devices and the devices' perceived helpfulness. Correlations between PROs and adherence to using the study devices in addition to their perceived helpfulness were determined performing correlation analyses as well. To determine statistical significance, a significance level of .05, which corresponds to a 95% CI, was used for all analyses. If a subject were readmitted during the monitoring period, to avoid partial data on the day of their admission, adherence rates were calculated only up to the day before the readmission. Any surveys completed by subjects whose first readmission occurred before the halfway mark of their next follow-up survey were not considered.

# Results

## Demographics

We evaluated 150 hospitalized heart failure patients between May 2018 and June 2019 for study eligibility. Of these 150 patients, 23 (15.3%) were discharged before they could be approached regarding the study, and 32 (21.3%) declined to participate. Another 69 (46.0%) patients did not meet the inclusion criteria, including 42 (28.0%) without a smartphone and eight (5.3%) who did not meet the minimum MDPQ score. In total, 20 (13.3%) heart failure patients were enrolled in the study.

The subjects' mean age was 65.3 years (SD 9.3; range 50–86). Half of the subjects were women, and 36.8% were African American (Table 2). Of the subjects, a high school degree was the highest level of education for 25.0%, whereas 35.0% had received a bachelor's degree or higher. The proportions of subjects whose families earned less than $50,000 (52.6%) and more

**Table 2. Demographics of the study population.**

| Characteristic | Value |
|---|---|
| Age (years; $n$ = 20), mean (SD) | 65.3 (9.3) |
| **Sex ($n$ = 20), $n$ (%)** | |
| Male | 10 (50.0) |
| Female | 10 (50.0) |
| **Hispanic or Spanish origin ($n$ = 20), $n$ (%)** | |
| No | 18 (90.0) |
| Yes | 2 (10.0) |
| **Race or Ethnicity ($n$ = 19), $n$ (%)** | |
| White | 12 (63.2) |
| Black or African American | 7 (36.8) |
| Asian | 0 (0.0) |
| American Indian or American Native | 0 (0.0) |
| Native Hawaiian or other Pacific Islander | 0 (0.0) |
| **Education ($n$ = 20), $n$ (%)** | |
| High school | 5 (25.0) |
| Some college, associate degree, or trade school | 8 (40.0) |
| Bachelor's degree | 4 (20.0) |
| Master's degree or above | 3 (15.0) |
| **Annual Income (US $; $n$ = 19), $n$ (%)** | |
| 0–25,000 | 4 (21.1) |
| 25,001–50,000 | 6 (31.6) |
| 50,001–75,000 | 2 (10.5) |
| 75,001 or more | 7 (36.8) |
| **New York Heart Association class ($n$ = 11), $n$ (%)** | |
| I | 0 (0.0) |
| II | 2 (18.2) |
| III | 5 (45.5) |
| IV | 4 (36.4) |
| **Ejection fraction ($n$ = 20), $n$ (%)** | |
| $\leq$40% | 13 (65.0) |
| 41%-49% | 0 (0.0) |
| $\geq$50% | 7 (35.0) |

than $50,000 (47.4%) were fairly similar. In regard to heart failure, 81.8% of the subjects were determined to be NYHA Class III or IV, and 65.0% had EFs less than 50%.

## Access to technology

In contrast to only seven subjects owning a tablet (36.8%), 17 subjects owned a smartphone (89.5%) (Table 3). All subjects had access to the internet through a wireless network, and the majority had additional internet access through a cellular network (75.0%). Most subjects had experience accessing the internet or their email account(s) (85.0%) and researching information about heart failure (60.0%). Fewer subjects had previously used apps on their smartphones to achieve health-related goals (52.6%), to make decisions about treatment (44.4%), and to ask a doctor new questions or to get a second opinion (47.4%).

## Patient-reported outcomes

The median MLHFQ score was 66.5 (IQR: 42.1–73.8) at baseline, corresponding to a poor HRQOL (Table 4), while the median KCCQ score (45.7, IQR: 35.7–58.6) at baseline suggested a fair HRQOL in regard to heart failure. The median SAQ score (57.4, IQR: 48.9–72.4) at baseline suggested a good HRQOL with respect to angina. Subjects had adequate ability to perform maintenance behaviors (median SCHFI score in maintenance of 70.0, IQR: 51.9–83.3 at baseline), but inadequate confidence level (median SCHFI score in confidence of 60.0, 45.0–75.0 at baseline). The median SCHFI score in management (66.7, IQR: 55.6–77.8) at baseline also revealed inadequate ability to manage heart failure for the 18 subjects who had experienced recent breathing complications or ankle swelling.

According to the median and IQR scores at baseline for the GDS-15 (4.0, IQR: 2.0–5.5), the LSNS-6 Family subscale (20.0, IQR: 15.0–21.0), and the LSNS-6 Friendships subscale (17.0,

**Table 3. Patient answers to the Health Information National Trends Survey.**

| Question | No | Yes |
|---|---|---|
| Do you ever go on-line to access the Internet or World Wide Web, or to send and receive e-mail? (*n* = 20) | 3 (15.0) | 17 (85.0) |
| When you use the Internet, do you ever access it through a regular dial-up telephone line? (*n* = 20) | 18 (90.0) | 2 (10.0) |
| When you use the Internet, do you ever access it through broadband such as DSL, cable or FiOS? (*n* = 20) | 6 (30.0) | 14 (70.0) |
| When you use the Internet, do you ever access it through acellular network (i.e., phone, 3G/4G)? (*n* = 20) | 5 (25.0) | 15 (75.0) |
| When you use the Internet, do you ever access it through a wireless network (Wi-Fi)? (*n* = 20) | 0 (0.0) | 20 (100.0) |
| In the past 12 months, have you used the Internet to look for heart failure information for yourself? | 6 (30.0) | 14 (70.0) |
| Do you own a tablet? (*n* = 19) | 7 (36.8) | 12 (63.2) |
| Do you own a smartphone? (*n* = 19) | 2 (10.5) | 17 (89.5) |
| Do you own a cell phone? (*n* = 19) | 3 (15.8) | 16 (84.2) |
| On your tablet or smartphone, do you have any software applications or "apps" related to health? (*n* = 20) | 5 (25.0) | 15 (75.0) |
| Have the apps on your smartphone or tablet related to health helped you achieve a health-related goal such as quitting smoking, losing weight, or increasing physical activity? (*n* = 19) | 9 (47.4) | 10 (52.6) |
| Have the apps on your smartphone or tablet related to health helped you make a decision about how to treat an illness or condition? (*n* = 18) | 10 (55.6) | 8 (44.4) |
| Have the apps on your smartphone or tablet related to health led you to ask a doctor new questions, or to get a second opinion from another doctor? (*n* = 19) | 10 (52.6) | 9 (47.4) |

**Table 4. Questionnaire scores at baseline.**

| Questionnaire | Median Score (IQR) |
|---|---|
| **Minnesota Living with Heart Failure Questionnaire** | |
| Physical (*n* = 20) | 28.5 (16.5–34.0) |
| Emotional (*n* = 20) | 11.0 (8.0–15.0) |
| Total (*n* = 20) | 66.5 (42.1–73.8) |
| **Self-Care of Heart Failure Index** | |
| Maintenance (*n* = 19) | 70.0 (51.9–83.3) |
| Management (*n* = 18) | 60.0 (45.0–75.0) |
| Confidence (*n* = 19) | 66.7 (55.6–77.8) |
| Geriatric Depression Scale (*n* = 20) | 4.0 (2.0–5.5) |
| **Lubben Social Network Scale** | |
| Family (*n* = 20) | 20.0 (15.0–21.0) |
| Friendships (*n* = 19) | 17.0 (14.0–19.0) |
| Seattle Angina Questionnaire (*n* = 17) | 57.4 (48.9–72.4) |
| Kansas City Cardiomyopathy Questionnaire (*n* = 19) | 45.7 (35.7–58.6) |
| **PROMIS Global Health** | |
| Physical (*n* = 17) | 37.4 (34.9–45.0) |
| Mental (*n* = 17) | 45.8 (41.1–53.4) |
| PROMIS Physical Function (*n* = 19) | 36.0 (30.3–39.3) |
| PROMIS Fatigue (*n* = 19) | 62.7 (57.0–69.0) |
| PROMIS Anxiety (*n* = 19) | 55.6 (48.8–60.7) |
| PROMIS Depression (*n* = 18) | 54.8 (49.0–58.9) |
| PROMIS Sleep Disturbance (*n* = 18) | 55.2 (54.3–61.7) |
| PROMIS Social Isolation (*n* = 18) | 46.8 (34.8–51.8) |

IQR: 14.0–19.0), most subjects were not depressed or at-risk for social isolation (Table 4). Similarly, median scores for the following PROMIS questionnaires concerning mental health at baseline were within the normal range: Global Mental Health subscale (45.8, IQR: 41.1–53.4), Anxiety (55.6, IQR: 48.8–60.7), Depression (54.8, IQR: 49.0–58.9), Sleep Disturbance (55.2, IQR: 54.3–61.7), and Social Isolation (46.8, IQR: 34.8–51.8). On the other hand, median scores for the PROMIS questionnaires concerning physical health at baseline revealed moderate to severe adverse health effects: Global Physical Health subscale (37.4, IQR: 34.9–45.0), Physical Function (36.0, IQR: 30.3–39.3), and Fatigue (62.7, IQR: 57.0–69.0).

Fewer subjects completed the SAQ and the management section of the SCHFI in their follow-up surveys because they no longer experienced the symptoms (chest pain, chest tightness, or angina, and trouble breathing or ankle swelling, respectively) that make them eligible to complete those questionnaires (S4 Table in S1 Appendix). Fig 1, which shows the average changes in subjects' questionnaire scores, indicates improvements in heart failure maintenance, heart failure management, angina, heart failure, physical function, fatigue, anxiety, depression, sleep disturbance and social isolation.

## Remote patient monitoring

Over the course of the study, one subject withdrew and three subjects expired, including one who made the transition to hospice. There were 10 different subjects who were readmitted to the hospital at least once and a total of 21 all-cause hospital readmissions. Two (9.5%) readmissions occurred within 30 days of discharge, while 11 (52.4%) occurred between 30 and 90 days and 8 (38.1%) between 90 and 180 days. Of the 21 hospital readmissions, four (19.0%) included heart failure in the admission diagnosis.

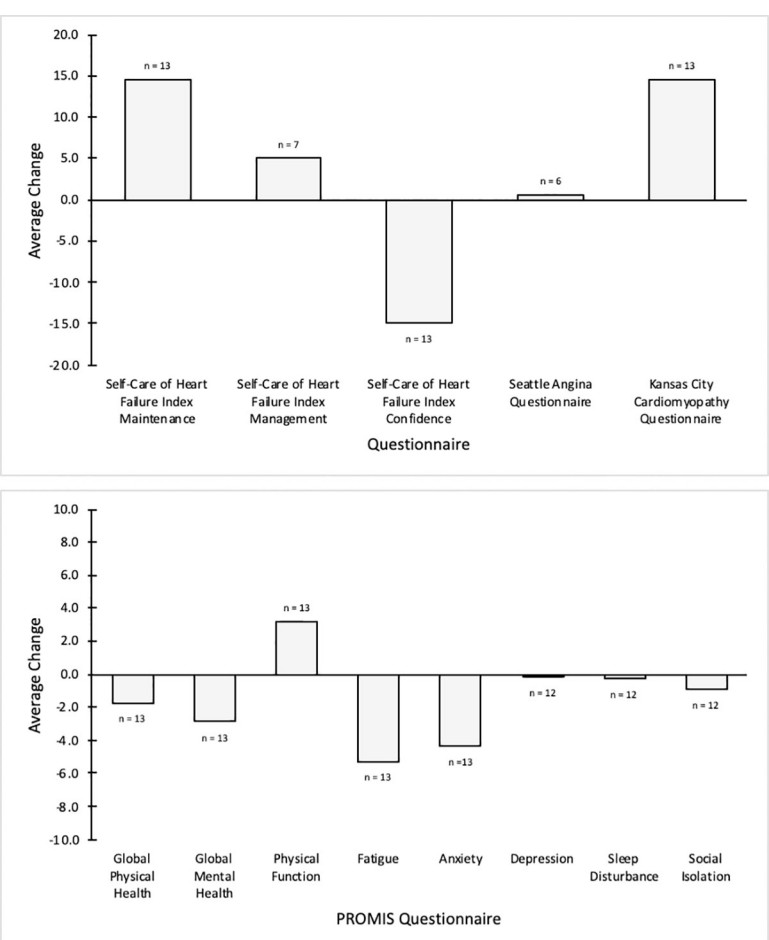

**Fig 1. Average changes in patient-reported outcomes.** For non-PROMIS questionnaires, a positive change indicates improvement in health status. A positive change also signifies improvement in health status for the following PROMIS questionnaires: Global Physical Health, Global Mental Health, and Physical Function. Conversely, a negative change is indicative of improvement in health status for the following PROMIS questionnaires: Fatigue, Anxiety, Depression, Sleep Disturbance, and Social Isolation.

Fig 2 illustrates the proportion of activity tracker (HR-hour and HR-minute), bathroom scale, and smart pill bottle usage across all subjects. The median usage percentage of the activity tracker was 79.1% for HR-hour and 75.4% for HR-minute. Usage percentages of the bathroom scale and smart pill bottle were 59.7%, and 2.8%, respectively.

## Correlations

HR-hour was selected for analysis of summary statistics and correlations. Device usage was not significantly correlated with collected subject characteristics (i.e., NYHA classification, EF, age, education, and annual income). On the other hand, HR-hour generated significant negative correlations with changes in SCHFI confidence subscale scores (r = -0.61; 95% CI: -0.09, -0.87), as well as changes in SAQ scores (r = -0.94; 95% CI: -0.30, -1.00). For bathroom scale usage, its negative correlation with follow-up SCHFI confidence subscale scores was the only significant result (r = -0.72; 95% CI: -0.30, -0.90). Smart pill bottle usage did not produce any significant correlations with PROs or their changes.

Perceived helpfulness of the activity tracker (S5 Table in S1 Appendix) produced significant positive correlations with HR-hour (r = 0.79; 95% CI: 0.41, 0.93) and follow-up SCHFI

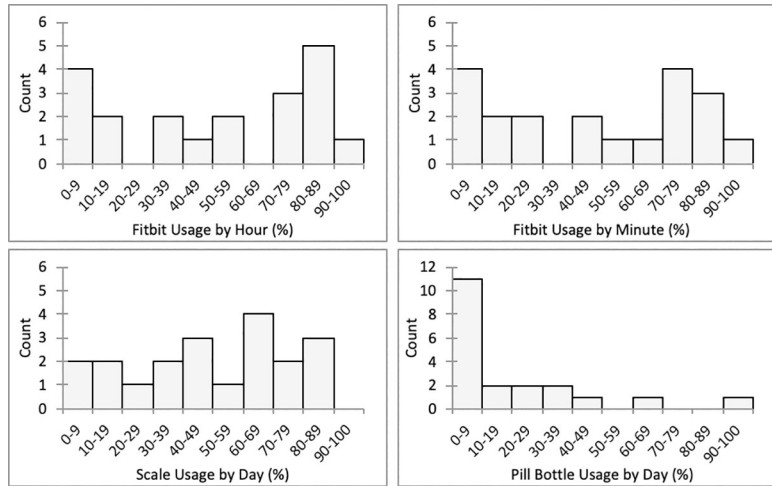

**Fig 2. Histograms of activity tracker, bathroom scale, and smart pill bottle usage.** Median (IQR) usage percentages were 79.1% (27.1%-90.6), 75.4% (23.4%-84.9%), 59.7% (24.6%-79.4%), and 2.8% (0.0%-54.3%) for HR-hour, HR-minute, bathroom scale, and smart pill bottle, respectively.

management subscale scores (r = 0.75; 95% CI: 0.09, 0.95). Additionally, it formed a significant negative correlation with subject age (r = -0.77; 95% CI: -0.37, -0.93). Subject age and the smart pill bottle's perceived helpfulness also negatively correlated with significance (r = -0.72; 95% CI: -0.31, -0.91). Perceived helpfulness of the bathroom scale and its usage positively correlated with significance (r = 0.54; 95% CI: 0.01, 0.83), while it generated a significant negative correlation with EFs (r = -0.59; 95% CI: -0.09, -0.85).

## Discussion

### Principal findings

Smartphone ownership and patient discharge were some of the biggest factors that adversely affected our enrollment rate. Results show that 42 heart failure patients did not own a smartphone, and 23 were discharged before they could be approached regarding the study. These two factors accounted for 50% (65/130) of the patients who were excluded from the study.

The median usage percentage of the activity tracker was 79.1%, and 60.0% of the subjects wore the device for at least 70% of the hours (Fig 1). When asked about their opinion of the activity tracker (S3 Table in S1 Appendix), most subjects alluded to the usefulness of the step count and heart rate tracking features. The notification and community features were mentioned as well but not as extensively. The bathroom scale's median usage percentage was lower at 59.7%, but nearly half (45.0%) of the subjects surpassed the usage rate of 70% (Fig 1). Despite using the bathroom scale at a lower rate than the activity tracker, the majority of subjects found it easy to incorporate into their lives and tried to use it every day. Conversely, subjects were much less adherent to using the smart pill bottle, as over half (55.0%) of the subjects used the device less than 10% of the days including seven (35.0%) who did not use the device at all (Fig 1). The most common feedback subjects provided regarding the pill bottle was that its medication reminders were helpful, but that their pill box was preferable to manage their medications. Lastly, the 30-day, 90-day, and 180-day follow-up surveys had completion rates of 50%, 55%, and 65%, respectively.

Common among all three study devices was a decrease in usage over the course of the study (S6 Table in S1 Appendix). This is consistent with previous observations in eHealth studies,

which have observed the general trend that patients may gradually lose interest in or become burdened by the study [39]. In a similar study conducted with chronically ill patients and telemonitoring devices, including a Fitbit activity tracker, patients used only the devices of interest to them after feeling overwhelmed by having multiple devices [40]. Despite study personnel's efforts to monitor and improve adherence by issuing reminders to the subjects in our study, many subjects chose not to continue using select devices.

Usage of the activity tracker by subjects in this study significantly correlated with changes in SCHFI confidence subscale scores and changes in SAQ scores from baseline with significance. These negative correlations may suggest that subjects who became less confident in their self-care or began to experience worse chest pain, chest tightness, or angina over the course of the study used the activity tracker more. The significant negative correlation between the changes in SAQ scores and bathroom scale usage may indicate that those who began experiencing worse chest pain, chest tightness, or angina used the bathroom scale more as well. Subjects with lower EFs found the bathroom scale more helpful, which may be related to physicians' frequent recommendation of daily weight monitoring as a part of heart failure self-management [41]. Furthermore, the subjects' usage and liking for the bathroom scale suggest the potential for health professionals to prevent fluid volume overload in heart failure patients by remotely monitoring their weight and utilizing predictive algorithms that predict hospitalizations [42].

## Limitations and future directions

This pilot study had a small sample of 20 patients and was confined to those admitted as an inpatient or for observation at a university-based health system. It was also restricted to heart failure patients over the age of 50. Though more than a third (36.8%) of the subjects were African American, the remaining (63.2%) were white (Table 2). Consequently, the results may not be applicable to the general population with heart failure.

Surveys were only available in English, and thus literacy in English was required. Future studies should include translated versions of the surveys in other languages. Though subjects did not have an incentive to give false answers, it is possible that an individual became fatigued and answered questions without giving them real consideration. Having study coordinators personally administer these surveys should reduce this risk, but this could be considered a limitation. While some questionnaires asked health-related questions that specified a time frame (e.g., previous 2 or 4 weeks), most of them concerned the subjects' status at the time each survey was administered. As a result, the daily variability in PROs were not accounted for due to the intervals of the surveys.

The number of medications each patient was taking and specific information about those medications were not considered as only one smart pill bottle was issued to each patient. Data collection may have been impacted by incorrect usage of the devices. For instance, subjects may not have worn the activity tracker properly, synced the activity tracker regularly, or stood on the bathroom scale long enough for the weight measurement to transmit.

## Conclusions

The low enrollment rate suggests difficulties in recruiting heart failure patients prior to their hospital discharge. Though the continued rise of smartphone ownership among older adults may improve the enrollment rate [43], adjustments to the inclusion criteria should be considered. Relaxing the smartphone ownership requirement by allowing one's caretaker to sync the activity tracker with his/her smartphone may circumvent cases in which the patient does not own a smartphone. Enrollment rate may also be improved by recruiting patients outside of the hospital (e.g., at outpatient appointment or by phone).

Considering the subjects' usage of and feedback regarding the activity tracker and bathroom scale, including those devices in a remote monitoring regimen may be feasible. Since Fitbit and BodyTrace provide data access in real time, patients may be able to use the data and predictive algorithms to monitor their physical health, in addition to health professionals remotely monitoring them. Moreover, given their non-invasive nature, low cost, and wide availability, the activity tracker and bathroom scale may have the potential to become a preferred alternative to more invasive and costly interventions, such as implantable pulmonary artery monitors [44]. On the other hand, pulmonary artery monitors may be preferable for some patients based on the relative predictive accuracy of the two approaches. A larger follow-up study is necessary to demonstrate the predictive value of these remote monitoring devices.

While the activity tracker and bathroom scale were positively received, the smart pill bottle was generally not useful and the follow-up surveys had low completion rates. For populations with complex medication regimens, monitoring medication usage is challenging and likely cannot be accomplished with a single pill bottle. There is a critical need for remote sensing technologies to capture medication adherence information. To improve the completion rates of follow-up surveys, administering abbreviated versions of the surveys more regularly and through an easier interface, such as a smartphone app, should be considered.

## Supporting information

**S1 Appendix.**
(DOCX)

**S1 Fig. Average changes in patient-reported outcomes 30 days after discharge.** For non-PROMIS questionnaires, a positive change indicates improvement in health status. A positive change also signifies improvement in health status for the following PROMIS questionnaires: Global Physical Health, Global Mental Health, and Physical Function. Conversely, a negative change is indicative of improvement in health status for the following PROMIS questionnaires: Fatigue, Anxiety, Depression, Sleep Disturbance, and Social Isolation.
(TIF)

**S2 Fig. Average changes in patient-reported outcomes 90 and 180 days after discharge.** For non-PROMIS questionnaires, a positive change indicates improvement in health status. A positive change also signifies improvement in health status for the following PROMIS questionnaires: Global Physical Health, Global Mental Health, and Physical Function. Conversely, a negative change is indicative of improvement in health status for the following PROMIS questionnaires: Fatigue, Anxiety, Depression, Sleep Disturbance, and Social Isolation.
(TIF)

**S1 Dataset.**
(XLSX)

**S2 Dataset.**
(XLSX)

**S3 Dataset.**
(XLSX)

## Author Contributions

**Conceptualization:** William Speier, Gregg C. Fonarow, Michael K. Ong, Corey W. Arnold.

**Data curation:** Albert Sohn, Esther Lan, Kymberly Aoki.

**Formal analysis:** Albert Sohn.

**Funding acquisition:** Corey W. Arnold.

**Investigation:** Albert Sohn, Esther Lan, Kymberly Aoki.

**Methodology:** William Speier, Corey W. Arnold.

**Project administration:** Esther Lan, Kymberly Aoki.

**Supervision:** William Speier, Gregg C. Fonarow, Michael K. Ong, Corey W. Arnold.

**Visualization:** Albert Sohn.

**Writing – original draft:** Albert Sohn.

**Writing – review & editing:** Albert Sohn, William Speier.

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
