## [Decision Letter · Decision Letter 0]

5 Aug 2020

PONE-D-20-21712

Integrating Remote Monitoring into Heart Failure Patients' Care Regimen: A Pilot Study

PLOS ONE

Dear Dr. Sohn,

Thank you for submitting your manuscript to PLOS ONE. After careful consideration, we feel that it has merit but does not fully meet PLOS ONE’s publication criteria as it currently stands. Therefore, we invite you to submit a revised version of the manuscript that addresses the points raised during the review process.

Thank you for your submission on this important but understudied topic. There is clearly a need to improve at-home treatment monitoring and compliance in order reduce HF readmissions. The pilot results presented herein provide a great deal of interesting information in this area, but substantial revisions are required in order to make the results more digestible for readers such that this paper can serve as a starting point for additional study.

In addition to the concerns raised by the reviewers, please consider addressing the following issues as well:

-the lack of an *a priori* definition of "pragmatic feasibility" is a significant limitation

-the statistical analysis section does not provide enough detail for the reader to fully understand exactly what was done. for example, line 239 mentions "regression analyses", but it is not clear if these are univariate or mulitivariable. also, what is the definition of "adherence rate"?

-given the numerous comparisons/hypotheses tested, i highly suspect alpha-inflation and a type 1 error. i would suggest correcting for multiple comparisons or simply reporting measures of uncertainly (eg, 95%CIs) rather than p-values, especially given the small sample size and thus low power.

-please include explicit details on the amount of missing data. the reference (35) for the imputation method reports that reliability declines when missingness is >20%--do the instruments with imputed variables meet this requirement?  ref 35 also refers to imputation of Likert scale data--do all of the surveys use these?

-with so many surveys to complete, is there any concern about accuracy for individual responses? what was the schedule on which they were asked to complete them (eg, all on the same day?). were participants compensated?

We look forward to receiving your revised manuscript.

Kind regards,

Robert Ehrman, MD, MS

Academic Editor

PLOS ONE

Journal Requirements:

"This work was supported by the National Institutes of Health (https://www.nih.gov) and the National Heart, Lung, and Blood Institute (https://www.nhlbi.nih.gov) under grant R56HL135425 awarded to C.W.A., PhD. The funders had no role in study design, data collection and analysis, decision to publish, or preparation of the manuscript."

Reviewers' comments:

Reviewer's Responses to Questions

**Comments to the Author**

1. Is the manuscript technically sound, and do the data support the conclusions?

Reviewer #1: No

Reviewer #2: Partly

2. Has the statistical analysis been performed appropriately and rigorously? 

Reviewer #1: I Don't Know

Reviewer #2: Yes

3. Have the authors made all data underlying the findings in their manuscript fully available?

Reviewer #1: Yes

Reviewer #2: Yes

4. Is the manuscript presented in an intelligible fashion and written in standard English?

Reviewer #1: Yes

Reviewer #2: Yes

5. Review Comments to the Author

Reviewer #1: Thank you for the opportunity to review this pilot study. I have a few specific observations and questions for the authors to consider:

- Weight gain/loss is a complicated, and potentially problematic, endpoint for assessing efficacy of chronic heart failure care. This is particularly true if the overall goal, as appears to be the case in this study, is to prevent episodes of acute heart failure (AHF) requiring hospitalization. The understanding of the role volume overload plays in AHF has changed dramatically in recent years, and it's now recognized that volume overload (and its surrogate of weight gain) is wholly insufficient to explain many AHF presentations. I would refer the authors to carefully review an article from 2017 by Fudim et al (https://www.ncbi.nlm.nih.gov/pmc/articles/PMC5586477/) which serves as an excellent introduction to the new paradigm of understanding Acute on chronic presentations of heart failure. A few particularly pertinent points to consider, however, are listed here: 1. Weight gain is poorly correlated with need for AHF hospitalization in patients with chronic heart failure, and roughly 50% of patients gain an insignificant amount of weight before hospitalization overall. In fact, the sensitivity for weight gain in predicting hospitalization has been estimated as low as 9% (though with specificity as high as 98%). 2. Up to a third of AHF patients are normovolemic or hypovolemic on total body water analysis. 3. In the timeline of central circulation congestion, weight change is a late phenomenon, often occurring days to weeks after the development of increased cardiac filling pressures (see figure 1 in that Fudim article). With all that said, the heavy focus on weight change for the assessment of chronic HF self care is problematic in the current study. While weight change doesn't need to be ignored (after all, as mentioned the specificity's 98%), it is insufficient to be the centerpiece of a HF self-care program. I would recommend the authors address this more in their discussion, and also consider other components (see later comments) that would be important for building a self-care program for HF patients that more comprehensively addresses the current understanding of the reasons for AHF episodes among those with chronic HF.

- Medication non-adherence is also frequently overestimated by clinicians as the reason for a particular HF hospitalization (Hollenberg 2019, ACC Expert Consensus Decision Pathway on Risk Assessment, Management, and Clinical Trajectory of Patients Hospitalized With Heart Failure). As a standalone cause of AHF episodes, medication or diet non-adherence represents a minority of cases (see Dr. Fonarow's report on AHF precipitants in OPTIMIZE-HF from 2008 in JAMA Internal Medicine). Most patients have other precipitants or a mix of causes for their hospitalizations (also see the results of the international GREAT registry, Arrigo et al., which recently replicated Dr. Fonarow's work on AHF precipitants in an international AHF population). The authors should comment on this in their discussion and limitations.

- With the above bullet point acknowledged, it is ALSO true that non-adherence to guideline directed medical therapy (GDMT) for chronic HF is established in numerous studies to portend adverse outcomes including hospitalization. This is, however, a more nuanced issue than the current study examines. First, it is specific medications for specific patients that matter. A patient with HFrEF not taking their beta blocker or ACEi, for example, has a significantly different impact on outcomes than a patient with HFpEF (the latter of whom for which no significant benefit of these meds have been shown). From what I can tell, the authors did not account for this or comment on this in the manuscript. Second, It is unclear to me how medications were distributed with the smart pill bottles - e.g. did patients get a smart pill bottle for every medication they took? Cardiovascular meds only? HF GDMT meds only? How many medications were each patient taking and how did that relate to the observed low rate of compliance with the smart pill bottle (i.e. were patients with >10 meds less likely to use the bottle?)?

- Could the authors comment in their discussion on how a monitoring program such as the one they pilot here could contrast with implantable pulmonary artery monitors, given the latter's recent growth? Pulmonary artery monitors are used in many ways to accomplish the same goal as the program here (to identify changes in a patient's chronic HF that may indicate increasing congestion and decompensation) and have arguably been relatively effective at this. Moreover, they do not require as much patient burden in reporting compliance as the monitoring program here - it would seem that the piloted mobile interventions here ask a lot of the patient. Of course, the tradeoff is that such monitors are invasive, but a discussion of the ways in which non-invasive mobile interventions would (and would not) be an improvement over this alternative strategy is likely warranted in the discussion.

- The authors compiled a staggering number of different surveys into a single repeated measures survey assessment. The amount of questionnaires included versus the number of survey items in which a significant difference was found raises questions about signal vs. noise - i.e. did patient scores for the SAQ and the confidence items on the SCHFI (but not the other components of SCHFI) differ because the intervention's value was specifically reflected in the targets of these survey items, or was the difference in these items (vs. others where no difference was noted) observed simply because such a large number of comparisons were undertaken? This is a key methodological limitation, even in a pilot study, and needs further discussion.

- The authors also should detail more clearly the repeated measures regression methodology used - saying "regression analysis" here is insufficient. Additionally, as far as I can tell, the methods as described would mean that at least 4 separate repeated measures regressions (one for each adherence variable) were performed. If all of the "PRO" measures and patient characteristics were used for each of these regressions then this would mean a staggering amount of covariates were included in each model (>15?) despite a sample size of 20. This would mean a high chance of problems such as collinearity and overfitting. Conversely, if each regression included a single PRO measure and all the patient characteristics this would still be a large number of covariates for the sample size, and would mean that a few dozen regressions were performed. In this latter case, overfitting and/or co-linearity are still possible problems (and need commenting on), and additionally the more general problem of a massive number of comparisons becomes even greater.

- Pursuant to the previous point, the analysis and results are overall generally hard to follow, in no small part due to how many analyses are reported or inferred and how little methodological information is given. Granted that pilot research is exploratory by nature, with a sample of 20 patients it is truly difficult to draw helpful conclusions for next steps/future work from the way the methods and results section is currently written. The authors should consider paring things down significantly and focus on just a few relationships which are felt to signal an important basis for future study.

- I am struggling with the authors' conclusions. The primary conclusion seems to be that the monitoring program is feasible. However, I do not think I can draw the same conclusion given that only 20 of 150 patients were able to be enrolled over 1 year, followup was only 50% successful, and a major component of the monitoring (the pill bottle) had very low adherence and was poorly received. Additionally, the amalgamation of numerous survey instruments, the choices/rationale for each of which I can not seem to discern beyond all being generally related to HF, makes it difficult for me to say this is a feasible program. The observation that patients did not do well with the pill bottles, likely due to the complex nature of medication adherence (as the authors do mention), is interesting but again limited by the lack of description of how this program was administered (as detailed above) and the small sample size. Overall, the results as currently presented would generally lead me to draw the opposite conclusion - that this monitoring program struggled with feasibility and was somewhat unwieldy. Could the authors perhaps address this further, specifically adding more justification for the conclusion that the results show feasibility of the program when such a small proportion of screened patients were enrolled and follow-up rates among those enrolled was so low?

- It is noted that the senior author's NHLBI grant R56HL135425 is a larger study with some similar aims, but focused on developing a predictive algorithm from some of the monitoring methods used here. In the current manuscript I cannot tell - was the presently reported work the pilot study for the senior author's R56? If so, please comment on this in the manuscript and consider revising the discussion and conclusions to more specifically assess how the pilot results informed the larger NHLBI study. If not, are the results presented here actually the main results of that R56? If this is the case, I again would question that the current report shows the feasibility of such a mobile health program since the publicly available project description for the R56 says enrollment was planned to be 500 patients. Either way, a discussion of how these pilot results affect future research efforts is needed, as currently there is little such reflection, and meeting a 500 patient target with the 20 patient enrollment seen here raises several questions about feasibility and applicability.

Overall, I applaud the authors for taking on an important general aim (identifying better ways to track outpatient HF care and prevent AHF rehospitalizations). With that said, the small sample, low response rate, and relative disorganization of the results presented here make the report very difficult to interpret. I would like to see, in particular, a more focused discussion on what this pilot study means for the next steps in research. With that said, the small sample size may preclude such a discussion.

To my eyes the results would generally suggest the opposite of the authors' conclusion (that the mobile health program is feasible) but I do struggle with this assessment given the many sources of uncertainty in the text discussed above. Further justification and clarification of the manuscript towards this end would be helpful and, in my view, necessary for publication.

Finally, I would add that if the general conclusion of the study is the converse of how I have interpreted the authors' discussion - i.e. that this screening program was not feasible - this in and of itself would still be an interesting and useful finding possibly worthy of publication (especially in the context of why feasibility may have suffered). However, the discussion seems to focus more on the results as a stepping stone to future research (yet are unclear as to what that research would/should/could be).

Reviewer #2: Thank you for allowing me the opportunity to review this interesting manuscript on a critically important, though sorely understudied area in heart failure research. The paper has some very important insights and observations about a unique monitoring modality for ambulatory heart failure patients post discharge from the hospital. The strengths of the paper include the fairly lengthy period of study for the patients (180 days), the longitudinal design and the use of a FitBit by the patients. One of the major weaknesses in my opinion is that enormous amount of data on multiple devices and questionnaires that is all packaged together in a single manuscript. The desire to simply get all of this potentially important data into publication most likely led to this decision to package everything together, but to me this seems like 3 distinct interventions that are each worthy of study independently. Below please find some recommendations should you choose to revise the current manuscript:

1. The objective statement talks about examining the "pragmatic feasibility of the approach", but the methods of the paper do not specifically outline the way in which feasibility will be studied or established. Some of the components of feasibility are approached in the design of the study (acceptability, practicality, limited-efficacy testing) but no explicit mention of the components of feasibility are apparent in the design of the study. The other areas of feasibility such as implementation, adaptation and expansion are not a part of the design whatsoever. I would suggest revising the objective statement to more closely align with the research that was conducted. Ref: Am J Prev Med. 2009 May ; 36(5): 452–457. doi:10.1016/j.amepre.2009.02.002.

2. Limitations should emphasize the large number of patients that were eligible for enrollment but were excluded for a variety of reasons (especially the 42 patients that were not enrolled b/c they did not have a smart phone) as well as the lack of a comparison group. A pure control group would not have been practical, but one interesting approach for a comparison would be to look at the outcomes of the study cohort and compare them to the outcomes of the cohort that was unable to be enrolled b/c of a lack of a smartphone. Or even to compare them to a more general HF cohort from the institution.

3. I am not an expert on survey methodology but am unclear on the appropriateness of using averages of other questions from a questionnaire to fill in the blanks for unanswered questions. Would recommend having this specific technique examined further.

4. Focusing on the FitBit data in this manuscript seems to me to be a more focused approach that would remove a lot of the data clutter the paper suffers from b/c of the multiple devices and myriad analyses that were examined and performed. A revised objective statement could focus more strongly on the FitBit data.

5. The statements in the Discussion section about the negative correlations b/w the SCHFI and SAW score suggesting that subjects who became less confident in their self-care or began to experience worsening symptoms used the activity tracker more (and the bathroom scale) seems to be a bit of a reach based on the data presented. The intervals of the surveys were sufficiently far apart that it is hard to account for the daily variability in how a chronic HF might be feeling based on the data that was collected.

6. Lines 451-4 seems like a bit of a reach given the infrequent intervals of questionnaire completion and lack of adherence with questionnaire protocol overall

7. I agree 100% with the final sentence.

6. PLOS authors have the option to publish the peer review history of their article (what does this mean?). If published, this will include your full peer review and any attached files.

Reviewer #1: **Yes: **Nicholas Harrison

Reviewer #2: No

---

## [Author Response · Author response to Decision Letter 0]

5 Oct 2020

Editor: Robert Ehrman, MD, MS

1. the lack of an a priori definition of "pragmatic feasibility" is a significant limitation

Thank you for the feedback. We understand that we may have caused confusion and created a limitation by not presenting a definition of pragmatic feasibility. There were two main factors we wished to learn more about in our pilot: 1) is it feasible to enroll patients, and 2) once enrolled, how adherent would patients be. We have clarified this in the manuscript and removed mention of “pragmatic feasibility,” which could be subjective by context, varying across different individuals and organizations. Our intention with this paper is to report on the results of our pilot, rather than make a claim that our approach is generally “pragmatically feasible.”

2. the statistical analysis section does not provide enough detail for the reader to fully understand exactly what was done. for example, line 239 mentions "regression analyses", but it is not clear if these are univariate or mulitivariable. also, what is the definition of "adherence rate"?

We apologize for the confusion. The analyses we performed were correlation analyses, not regression analyses as stated in our initial submission. In regard to adherence rate, there are two adherence methods listed for the activity tracker (HR-hour and HR-minute), along with one for each of the bathroom scale and smart pill bottle. HR-hour is the number of hours in the study that each participant had a heartrate recording divided by the total number of hours in the study. Similarly, HR-minute is the number of minutes in the study that each participant had a heartrate recording divided by the total number of minutes in the study. For both the bathroom scale and smart pill bottle, adherence rate is the number of days in the study that each participant had data recorded on the device divided by the total number of days in the study. We have revised the section “Statistical analysis” to clarify these points.

3. given the numerous comparisons/hypotheses tested, i highly suspect alpha-inflation and a type 1 error. i would suggest correcting for multiple comparisons or simply reporting measures of uncertainly (eg, 95%CIs) rather than p-values, especially given the small sample size and thus low power.

Thank you for the suggestion. In response to this comment and consideration of the small sample size, we have removed p-values and replaced them with 95% CIs.

4. please include explicit details on the amount of missing data. the reference (35) for the imputation method reports that reliability declines when missingness is >20%--do the instruments with imputed variables meet this requirement? ref 35 also refers to imputation of Likert scale data--do all of the surveys use these?

We thank the reviewer for this comment. Only questionnaires that used Likert scales followed this method of imputation, and questionnaires missing >20% of the items were omitted from analysis. This discussion has been added to the section “Statistical analysis.” 

5. with so many surveys to complete, is there any concern about accuracy for individual responses? what was the schedule on which they were asked to complete them (eg, all on the same day?). were participants compensated?

Each of the surveys was administered by a study coordinator. Participants were asked to complete all questionnaires on each survey in one sitting (baseline) or call (30-day, 90-day, and 180-day). There was no direct compensation for completing follow-up surveys and no penalty for not taking them; study participants were not paid but were allowed to keep the study devices, which they were given at the beginning of the study after completing the baseline survey in person with a study coordinator. 

We do not have a robust method to evaluate the accuracy of responses. Even though subjects did not have an incentive to give false answers, it is possible that an individual became fatigued and answered questions without giving them real consideration. Having study coordinators personally administer these surveys should reduce this risk, but it is still a possibility. This could be considered a limitation of our study.

Reviewer 1: Nicholas Harrison

1. Weight gain/loss is a complicated, and potentially problematic, endpoint for assessing efficacy of chronic heart failure care. This is particularly true if the overall goal, as appears to be the case in this study, is to prevent episodes of acute heart failure (AHF) requiring hospitalization. The understanding of the role volume overload plays in AHF has changed dramatically in recent years, and it's now recognized that volume overload (and its surrogate of weight gain) is wholly insufficient to explain many AHF presentations. I would refer the authors to carefully review an article from 2017 by Fudim et al (https://www.ncbi.nlm.nih.gov/pmc/articles/PMC5586477/) which serves as an excellent introduction to the new paradigm of understanding Acute on chronic presentations of heart failure. A few particularly pertinent points to consider, however, are listed here: 1. Weight gain is poorly correlated with need for AHF hospitalization in patients with chronic heart failure, and roughly 50% of patients gain an insignificant amount of weight before hospitalization overall. In fact, the sensitivity for weight gain in predicting hospitalization has been estimated as low as 9% (though with specificity as high as 98%). 2. Up to a third of AHF patients are normovolemic or hypovolemic on total body water analysis. 3. In the timeline of central circulation congestion, weight change is a late phenomenon, often occurring days to weeks after the development of increased cardiac filling pressures (see figure 1 in that Fudim article). With all that said, the heavy focus on weight change for the assessment of chronic HF self care is problematic in the current study. While weight change doesn't need to be ignored (after all, as mentioned the specificity's 98%), it is insufficient to be the centerpiece of a HF self-care program. I would recommend the authors address this more in their discussion, and also consider other components (see later comments) that would be important for building a self-care program for HF patients that more comprehensively addresses the current understanding of the reasons for AHF episodes among those with chronic HF.

We agree that monitoring weight alone or making weight change the centerpiece of a heart failure self-care program would not be comprehensive. We did not intend for weight monitoring to be the centerpiece of this remote monitoring protocol, but rather a component of a multimodal system including heartrate, activity, and medication adherence. While weight change may be a poor predictor of rehospitalization on its own, it could be a valuable feature for some patients, and may offer signal when used in conjunction with other data sources (which is supported by the high specificity in the study the reviewer mentioned).

Instead of addressing this in the discussion, we think it may be better addressed in the background section, especially since this knowledge influenced our study design and selection of devices that could track other aspects of self-care (i.e., activity tracker and smart pill bottle). In response to this comment, we have included a statement that states “weight change or medication non-adherence alone, for example, may not correlate with hospitalizations” and cited the article referred by the reviewer. Furthermore, to clarify our overall goal and purpose of including a bathroom scale in our study, we have revised the objective section. It now indicates that we intended to demonstrate the potential of different remote monitoring devices that can provide information, which could be combined in a predictive algorithm.

2. Medication non-adherence is also frequently overestimated by clinicians as the reason for a particular HF hospitalization (Hollenberg 2019, ACC Expert Consensus Decision Pathway on Risk Assessment, Management, and Clinical Trajectory of Patients Hospitalized With Heart Failure). As a standalone cause of AHF episodes, medication or diet non-adherence represents a minority of cases (see Dr. Fonarow's report on AHF precipitants in OPTIMIZE-HF from 2008 in JAMA Internal Medicine). Most patients have other precipitants or a mix of causes for their hospitalizations (also see the results of the international GREAT registry, Arrigo et al., which recently replicated Dr. Fonarow's work on AHF precipitants in an international AHF population). The authors should comment on this in their discussion and limitations.

We thank the reviewer for this insight as well and agree that simply tracking medication non-adherence is insufficient to prevent hospitalization. We would like to ensure that this knowledge was considered while designing the study. Related to the above response, we understand that medication non-adherence may not be a good predictor in isolation, but it could be helpful as a feature in a multimodal remote monitoring system. As the reviewer states, most patients have a mix of causes for hospitalization, so we aimed to combine information from as many sources as possible, which is why we are studying several monitoring devices. In response to this comment, we have cited the article referred by the reviewer and added this discussion in the section “Background” and “Objective” under “Introduction,” since we think it may be better addressed in those sections.

3. With the above bullet point acknowledged, it is ALSO true that non-adherence to guideline directed medical therapy (GDMT) for chronic HF is established in numerous studies to portend adverse outcomes including hospitalization. This is, however, a more nuanced issue than the current study examines. First, it is specific medications for specific patients that matter. A patient with HFrEF not taking their beta blocker or ACEi, for example, has a significantly different impact on outcomes than a patient with HFpEF (the latter of whom for which no significant benefit of these meds have been shown). From what I can tell, the authors did not account for this or comment on this in the manuscript. Second, It is unclear to me how medications were distributed with the smart pill bottles - e.g. did patients get a smart pill bottle for every medication they took? Cardiovascular meds only? HF GDMT meds only? How many medications were each patient taking and how did that relate to the observed low rate of compliance with the smart pill bottle (i.e. were patients with >10 meds less likely to use the bottle?)?

Thank you for the feedback. We realize that not accounting for the specific medications patients were taking throughout their participation in the study is a limitation. The goal here was to simplify things as much as possible to see if patients would use a smart pill bottle in the simplest case (one bottle, regardless of medication type). If we were able to measure adherence with this method, it could be further stratified by number of medications or medication type. However, usage of this device was low, which prevented further analysis. In response to this comment, we have revised the section “Limitations and future directions” to indicate that we did not account for the number of medications or specific medications each patient was taking. By addressing this in the limitations, we think it will be a bridge to the conclusions section where we suggest, “for populations with complex medication regimens, monitoring medication usage is challenging and likely cannot be accomplished with a single pill bottle.”

4. Could the authors comment in their discussion on how a monitoring program such as the one they pilot here could contrast with implantable pulmonary artery monitors, given the latter's recent growth? Pulmonary artery monitors are used in many ways to accomplish the same goal as the program here (to identify changes in a patient's chronic HF that may indicate increasing congestion and decompensation) and have arguably been relatively effective at this. Moreover, they do not require as much patient burden in reporting compliance as the monitoring program here - it would seem that the piloted mobile interventions here ask a lot of the patient. Of course, the tradeoff is that such monitors are invasive, but a discussion of the ways in which non-invasive mobile interventions would (and would not) be an improvement over this alternative strategy is likely warranted in the discussion.

Thank you for this suggestion. The motivation for this remote monitoring regimen is that it is non-invasive, less costly, and more widely available than implantable pulmonary artery monitors. Additionally, this approach could promote self-care by allowing heart failure patients to monitor their own status, in addition to health professionals remotely monitoring them. On the other hand, pulmonary artery monitors may be preferable for some patients based on the relative predictive accuracy of the two approaches. A larger follow-up study is necessary to demonstrate the predictive value of these remote monitoring devices. This discussion has been added to the conclusions section.

5. The authors compiled a staggering number of different surveys into a single repeated measures survey assessment. The amount of questionnaires included versus the number of survey items in which a significant difference was found raises questions about signal vs. noise - i.e. did patient scores for the SAQ and the confidence items on the SCHFI (but not the other components of SCHFI) differ because the intervention's value was specifically reflected in the targets of these survey items, or was the difference in these items (vs. others where no difference was noted) observed simply because such a large number of comparisons were undertaken? This is a key methodological limitation, even in a pilot study, and needs further discussion.

We thank the reviewer for this comment and agree that the statistical analysis of the survey responses was inadequate. While there were many surveys administered, many of these questionnaires had responses that were highly correlated. In our initial submission, we attempted to give examples that were representative of the trends we saw, but we realize that too many were included, and their presentation was poorly organized and confusing. We have simplified the presentation of the survey results to focus on the most important findings which hopefully makes things clearer. As mentioned in our response to the editor, we have also removed p values and instead used confidence intervals because they are a more appropriate analysis of these results given the small sample size and large number of survey questions administered.

6. The authors also should detail more clearly the repeated measures regression methodology used - saying "regression analysis" here is insufficient. Additionally, as far as I can tell, the methods as described would mean that at least 4 separate repeated measures regressions (one for each adherence variable) were performed. If all of the "PRO" measures and patient characteristics were used for each of these regressions then this would mean a staggering amount of covariates were included in each model (>15?) despite a sample size of 20. This would mean a high chance of problems such as collinearity and overfitting. Conversely, if each regression included a single PRO measure and all the patient characteristics this would still be a large number of covariates for the sample size, and would mean that a few dozen regressions were performed. In this latter case, overfitting and/or co-linearity are still possible problems (and need commenting on), and additionally the more general problem of a massive number of comparisons becomes even greater.

We apologize for the confusion. The analyses we performed were correlation analyses, not regression analyses. We have revised the section “Statistical analysis” to clarify this. We also agree that multivariate analyses would lead to a large number of covariates. Due to this fact and the small sample size, we did not perform multivariate regression analyses. We think the results of correlation analyses still offer insight that may be useful for future study.

8. Pursuant to the previous point, the analysis and results are overall generally hard to follow, in no small part due to how many analyses are reported or inferred and how little methodological information is given. Granted that pilot research is exploratory by nature, with a sample of 20 patients it is truly difficult to draw helpful conclusions for next steps/future work from the way the methods and results section is currently written. The authors should consider paring things down significantly and focus on just a few relationships which are felt to signal an important basis for future study.

Thank you for the feedback. In response to this comment, we have considerably pared down the results to focus on a few relationships that we think are important for future study. We also reorganized the results into five sections: Demographics, Access to technology, Patient-reported outcomes, Remote patient monitoring, and Correlation.

9. I am struggling with the authors' conclusions. The primary conclusion seems to be that the monitoring program is feasible. However, I do not think I can draw the same conclusion given that only 20 of 150 patients were able to be enrolled over 1 year, followup was only 50% successful, and a major component of the monitoring (the pill bottle) had very low adherence and was poorly received. Additionally, the amalgamation of numerous survey instruments, the choices/rationale for each of which I can not seem to discern beyond all being generally related to HF, makes it difficult for me to say this is a feasible program. The observation that patients did not do well with the pill bottles, likely due to the complex nature of medication adherence (as the authors do mention), is interesting but again limited by the lack of description of how this program was administered (as detailed above) and the small sample size. Overall, the results as currently presented would generally lead me to draw the opposite conclusion - that this monitoring program struggled with feasibility and was somewhat unwieldy. Could the authors perhaps address this further, specifically adding more justification for the conclusion that the results show feasibility of the program when such a small proportion of screened patients were enrolled and follow-up rates among those enrolled was so low?

Thank you for the feedback. We understand it is reasonable to think that the remote monitoring regimen is potentially not feasible from the points that the reviewer mentions above. We think the low enrollment rate is in large part due to the patients being discharged before they could be approached (15.3%) and the requirements of smartphone ownership and internet access, since at least 28.0% of the potential subjects did not own a smartphone. We realize these are limitations but believe future enrollment may improve as smartphone ownership is increasing among older adults [1]. Additionally, recruiting patients outside of the hospital (e.g., at outpatient appointments or by phone) and relaxing these requirements may be possibilities for future studies. The requirement that the patient owns a smartphone may be removed by allowing one’s caretaker to sync the Fitbit with his/her smartphone. Finally, we believe that there is no “one shoe fits all” intervention for the complex spectrum of heart failure patients, and that the proposed regimen is possibly a good option for some patients, but certainly not all. This discussion has been added to the conclusions section.

In our initial submission, we had a general statement that the remote monitoring is feasible as the reviewer noted. We have revised this statement in the conclusion section to say that usage of two of the devices may be feasible (activity tracker and bathroom scale), while the other two are not in their current form (smart pill bottle and follow-up surveys). We elaborated on our discussion of the smart pill bottle’s limitation for populations with complex medication regimens in the limitations and conclusions sections. For the follow-up surveys, administering abbreviated versions of the surveys more regularly and through an easier interface, such as a smartphone app, may improve their completion rate. We have added this discussion to the section “Limitations and future directions.”

10. It is noted that the senior author's NHLBI grant R56HL135425 is a larger study with some similar aims, but focused on developing a predictive algorithm from some of the monitoring methods used here. In the current manuscript I cannot tell - was the presently reported work the pilot study for the senior author's R56? If so, please comment on this in the manuscript and consider revising the discussion and conclusions to more specifically assess how the pilot results informed the larger NHLBI study. If not, are the results presented here actually the main results of that R56? If this is the case, I again would question that the current report shows the feasibility of such a mobile health program since the publicly available project description for the R56 says enrollment was planned to be 500 patients. Either way, a discussion of how these pilot results affect future research efforts is needed, as currently there is little such reflection, and meeting a 500 patient target with the 20 patient enrollment seen here raises several questions about feasibility and applicability.

The R56 supported the pilot study we present in the manuscript and was a bridge to our R01 award that will recruit 500 patients. NIH RePORTER likely lists the aims of the R01 for the R56 given one feature of the R56 mechanism is to allow an investigator to further develop a research plan by acquiring pilot results. The overall goal of the R56 pilot study was to determine the feasibility of conducting a study that uses device data to predict hospital readmissions by demonstrating that subjects would use the devices and that the data could be applied in that way. This discussion has been added to the objective section. Given the study objective, we think usage of the activity tracker and bathroom scale may be feasible, whereas the smart pill bottle and follow-up surveys are not in their current forms. In response to this comment, as well as the reviewer’s ninth comment above, we have revised the conclusions section to explain that we think usage of the two aforementioned devices may be feasible and the limitations and conclusions section to describe the issues and future directions of the smart pill bottle and follow-up surveys.

11. Overall, I applaud the authors for taking on an important general aim (identifying better ways to track outpatient HF care and prevent AHF rehospitalizations). With that said, the small sample, low response rate, and relative disorganization of the results presented here make the report very difficult to interpret. I would like to see, in particular, a more focused discussion on what this pilot study means for the next steps in research. With that said, the small sample size may preclude such a discussion.

To my eyes the results would generally suggest the opposite of the authors' conclusion (that the mobile health program is feasible) but I do struggle with this assessment given the many sources of uncertainty in the text discussed above. Further justification and clarification of the manuscript towards this end would be helpful and, in my view, necessary for publication.

Finally, I would add that if the general conclusion of the study is the converse of how I have interpreted the authors' discussion - i.e. that this screening program was not feasible - this in and of itself would still be an interesting and useful finding possibly worthy of publication (especially in the context of why feasibility may have suffered). However, the discussion seems to focus more on the results as a stepping stone to future research (yet are unclear as to what that research would/should/could be).

Thank you for the feedback. We understand the reviewer’s concerns about the small sample and low response rate. However, despite the small sample and low response rate, we believe the results offer important insight into which aspects of the remote monitoring regimen (activity tracker and bathroom scale) may be feasible and which aspects (smart pill bottle and follow-up surveys) are not in their current forms. We revised the limitations and conclusions sections to specifically mention the feasibility of the activity tracker and bathroom scale to be used in future studies, in addition to the shortcomings of the smart pill bottle and follow-up surveys. We think that revising these sections to discuss each aspect of the remote monitoring regimen individually rather than as a whole will clarify and justify our conclusions about their feasibility and future directions. We also have revised the presentation of the survey results to focus on the most important findings, hopefully making the results easier to follow. 

Reviewer 2

1.The objective statement talks about examining the "pragmatic feasibility of the approach", but the methods of the paper do not specifically outline the way in which feasibility will be studied or established. Some of the components of feasibility are approached in the design of the study (acceptability, practicality, limited-efficacy testing) but no explicit mention of the components of feasibility are apparent in thse design of the study. The other areas of feasibility such as implementation, adaptation and expansion are not a part of the design whatsoever. I would suggest revising the objective statement to more closely align with the research that was conducted. Ref: Am J Prev Med. 2009 May ; 36(5): 452–457. doi:10.1016/j.amepre.2009.02.002.

Thank you for the suggestion. We sought to investigate two main factors in our pilot: 1) feasibility of enrolling patients, and 2) their adherence to a remote monitoring reigmen. We have clarified this in the manuscript and removed mention of “pragmatic feasibility,” which is a somewhat subjective notion that will vary across individuals and organizations. Our intention with this paper is to report on the results of our pilot, rather than make a claim that our approach is generally “pragmatically feasible.”

2. Limitations should emphasize the large number of patients that were eligible for enrollment but were excluded for a variety of reasons (especially the 42 patients that were not enrolled b/c they did not have a smart phone) as well as the lack of a comparison group. A pure control group would not have been practical, but one interesting approach for a comparison would be to look at the outcomes of the study cohort and compare them to the outcomes of the cohort that was unable to be enrolled b/c of a lack of a smartphone. Or even to compare them to a more general HF cohort from the institution

We thank the reviewer for the suggestion and agree that the limitations should mention the large number of patients who were excluded for various reasons, as well as those who were discharged or declined to participate. This discussion has been added to the section “Limitations and future directions.” In regard to the comment about a control group for comparison, we agree that it would be interesting but believe that our sample size is inadequate to draw significant conclusions. However, it is something we may explore in a larger future study, since the subjects’ usage of and feedback regarding the activity tracker and bathroom scale were promising.

3. I am not an expert on survey methodology but am unclear on the appropriateness of using averages of other questions from a questionnaire to fill in the blanks for unanswered questions. Would recommend having this specific technique examined further.

Thank you for the feedback. We have revised the statistical analysis section to explain that only questionnaires that used Likert scales followed this method of imputation and that questionnaires missing >20% of the items were not subject to this method, since reliability declines when missingness is >20% [2].

4. Focusing on the FitBit data in this manuscript seems to me to be a more focused approach that would remove a lot of the data clutter the paper suffers from b/c of the multiple devices and myriad analyses that were examined and performed. A revised objective statement could focus more strongly on the FitBit data.

Thank you for the suggestion. We apologize for the confusion resulting from the high number of comparisons and analyses included in the paper. We have considerably pared down the results section to focus on a few relationships that we think are important for future study. We also reorganized the results into five sections: Demographics, Access to technology, Patient-reported outcomes, Remote patient monitoring, and Correlations. Because the objective of this manuscript was to suggest the feasibility of the remote monitoring regimen by detailing the preliminary results, we decided not to focus more on one device over the others. Instead, we have restructured the results to more clearly demonstrate the value of each of the modalities used. We hope this addresses the reviewer’s concern.

5. The statements in the Discussion section about the negative correlations b/w the SCHFI and SAW score suggesting that subjects who became less confident in their self-care or began to experience worsening symptoms used the activity tracker more (and the bathroom scale) seems to be a bit of a reach based on the data presented. The intervals of the surveys were sufficiently far apart that it is hard to account for the daily variability in how a chronic HF might be feeling based on the data that was collected.

We thank the reviewer for the feedback. We understand that there may be daily variability in how heart failure affects each individual and the wording in our initial submission was stronger than we intended it to be. Instead of suggesting a general trend, we have revised the statements to state that it is restricted to our observations because the sample size is not sufficient to make conclusions about the general population. Additionally, we have revised the section “Limitations and future directions” to discuss the aforementioned limitation of the questionnaires. 

6. Lines 451-4 seems like a bit of a reach given the infrequent intervals of questionnaire completion and lack of adherence with questionnaire protocol overall

Thank you for the feedback. We understand that there are limitations from the survey frequency and completion rates. In response to this comment, as well as the one above, we have revised the limitations section and those lines to stress that further research is necessary to study this observation.

References

[1] Anderson M, Perrin A. Pew Research Center: Internet & Technology. 2017. [2019-04-26]. Technology use among seniors https://www.pewinternet.org/2017/05/17/technology-use-among-seniors/

[2] Downey RG, King C. Missing data in Likert ratings: a comparison of replacement methods. J Gen Psychol. 1998;125(2):175–91.

---

## [Decision Letter · Decision Letter 1]

29 Oct 2020

Integrating Remote Monitoring into Heart Failure Patients' Care Regimen: A Pilot Study

PONE-D-20-21712R1

Dear Dr. Sohn,

We’re pleased to inform you that your manuscript has been judged scientifically suitable for publication and will be formally accepted for publication once it meets all outstanding technical requirements.

Kind regards,

Robert Ehrman, MD, MS

Academic Editor

PLOS ONE

Additional Editor Comments (optional):

Reviewers' comments:

Reviewer's Responses to Questions

**Comments to the Author**

1. If the authors have adequately addressed your comments raised in a previous round of review and you feel that this manuscript is now acceptable for publication, you may indicate that here to bypass the “Comments to the Author” section, enter your conflict of interest statement in the “Confidential to Editor” section, and submit your "Accept" recommendation.

Reviewer #2: All comments have been addressed

2. Is the manuscript technically sound, and do the data support the conclusions?

Reviewer #2: Yes

3. Has the statistical analysis been performed appropriately and rigorously? 

Reviewer #2: Yes

4. Have the authors made all data underlying the findings in their manuscript fully available?

Reviewer #2: Yes

5. Is the manuscript presented in an intelligible fashion and written in standard English?

Reviewer #2: Yes

6. Review Comments to the Author

Reviewer #2: thank you for the opportunity to review the manuscript. All of the issues that i raised with the original version of the manuscript have been adequately addressed.

7. PLOS authors have the option to publish the peer review history of their article (what does this mean?). If published, this will include your full peer review and any attached files.

Reviewer #2: **Yes: **Mark Favot

---

## [Editor Report · Acceptance letter]

10 Nov 2020

PONE-D-20-21712R1 

Integrating remote monitoring into heart failure patients’ care regimen: A pilot study 

Dear Dr. Sohn:

I'm pleased to inform you that your manuscript has been deemed suitable for publication in PLOS ONE. Congratulations! Your manuscript is now with our production department. 

Kind regards, 

on behalf of

Dr. Robert Ehrman 

Academic Editor

PLOS ONE